# An Integrated Front-end Circuit Board for Air-Coupled CMUT Burst-Echo Imaging

**DOI:** 10.3390/s20216128

**Published:** 2020-10-28

**Authors:** Lei Ye, Jian Li, Hui Zhang, Dongmei Liang, Zhuochen Wang

**Affiliations:** State Key Laboratory of Precision Measurement Technology and Instrument, Tianjin University, Tianjin 300072, China; yelei15@tju.edu.cn (L.Y.); tjupipe@tju.edu.cn (J.L.); hzhang@tju.edu.cn (H.Z.); dm_liang@tju.edu.cn (D.L.)

**Keywords:** front-end circuit, air-coupled CMUT, burst-echo imaging, imaging system

## Abstract

To conduct burst-echo imaging with air-coupled capacitive micromachined ultrasonic transducers (CMUTs) using the same elements in transmission and reception, this work proposes a dedicated and integrated front-end circuit board design to build an imaging system. To the best of the authors’ knowledge, this is the first air-coupled CMUT burst-echo imaging using the same elements in transmission and reception. The reported front-end circuit board, controlled by field programmable gate array (FPGA), consisted of four parts: an on-board pulser, a bias-tee, a T/R switch and an amplifier. Working with our 217 kHz 16-element air-coupled CMUT array under 100 V DC bias, the front-end circuit board and imaging system could achieve 22.94 dB signal-to-noise ratio (SNR) in burst-echo imaging in air, which could represent the surface morphology and the three-dimensional form factor of the target. In addition, the burst-echo imaging range of our air-coupled CMUT imaging system, which could work between 52 and 273 mm, was discussed. This work suggests good potential for ultrasound imaging and gesture recognition applications.

## 1. Introduction

In recent years, air-coupled ultrasonic testing technology has played a significant role in the medical [1] and aerospace [2] industries, human–computer interaction (HCI) [3] and other fields, owing to the advantages of being noncontact and noninvasive [4]. Compared to traditional piezoelectric ultrasonic transducers, capacitive micromachined ultrasonic transducers (CMUTs) with wide bandwidth, good acoustic matching with air, easily fabricated high-density arrays and integration with front-end circuits are suitable for air-coupled ultrasonic applications [5,6,7,8].

Air-coupled CMUTs have been studied in a variety of applications for a long time. M. Kupnik et al. [9] designed an air-coupled CMUT that had its −6 dB bandwidth improved from 1% to 2.5% for ultrasonic transit-time detection. K. K. Park et al. [10] constructed 3D synthetic aperture imaging of two plates at distances of 350 and 400 mm using a pair of air-coupled CMUTs. S. Na et al. [11] designed CMUTs based on annular cell geometry with the ratio of average-to-maximum displacement (RAMD) of 0.52–0.58 for pitch–catch applications in air. P. Shanmugam et al. [12] used CMUTs to monitor binary gas mixtures by simultaneously sensing variations in ultrasonic velocity and ultrasonic attenuation. However, previous applications of air-coupled CMUTs did not use the same CMUT element to transmit and receive ultrasonic waves. Different transmission and reception elements create a deflection angle between the transmitting and receiving beams, which complicates beam path analysis. In addition, different transmission and reception elements also reduce the utilization ratio of CMUT elements, which conflicts with its merits of miniaturization and integration.

There are still some difficulties in transmitting and receiving with the same element in air-coupled CMUTs, leading to a low signal-to-noise ratio (SNR), which impedes imaging applications in air. The main reasons include low receiving sensitivity of CMUTs and noise introduced by the T/R switch in the receiving circuit, both greatly reducing the SNR of the echo signal.

In order to achieve air-coupled CMUT applications with the same transmission and reception elements, this paper reported a dedicated and integrated front-end circuit board controlled by a field-programmable gate array (FPGA). Using the designed front-end circuit board with 22.94 dB SNR, burst-echo imaging with our previously reported air-coupled CMUT [13], which could represent the surface morphology and the three-dimensional form factor of the target, was carried out. The imaging result demonstrated the feasibility of air-coupled CMUT imaging using the same transmission and reception elements, suggesting good potential in ultrasound imaging and gesture recognition applications in the future.

## 2. Circuit Design

In air-coupled CMUT imaging systems, the front circuit plays an important role in signal collecting and system integration. Some requirements should be considered when designing the front-end circuit for air-coupled CMUTs. First of all, since CMUTs require a DC bias to transmit and receive ultrasonic waves [14], the front-end circuit has to include a bias-tee. Secondly, in order to reduce the volume of the CMUT application system, it is necessary to design an integrated on-board pulser. Moreover, in burst-echo applications with the same transmission and reception elements, a T/R switch should be designed to avoid mutual interference between the transmitting and receiving channels. Furthermore, when designing an amplifier, it is necessary to ensure that the amplifier has sufficient gain, bandwidth and SNR.

As a result, the front-end circuit for air-coupled CMUTs consists of four parts: an on-board pulser, a bias-tee, a T/R switch and an amplifier, as shown in Figure 1.

### 2.1. On-Board Pulser

A switch-type high-voltage square-wave generation circuit was used to generate a bipolar square-wave burst (Figure 2). The low-voltage low-current square-wave burst generated by FPGA (xc7k325t-ffg900i, Xilinx, SAN Jose, CA, USA) was amplified by the MOSFET driver (MD1822, Microchip, Chandler, AZ, USA) from 3.3 to 10 V to drive high-speed PMOS/NMOS with sufficient voltage and current. Then, the PMOS (Q1, Q3) and NMOS (Q2, Q4), both using TC6320 (Microchip, Chandler, AZ, USA) in this design, generated a high-voltage (up to 200 Vpp) bipolar square-wave burst through the switch action. Using the characteristics of the capacitors, the rapid change of MOSFET’s gate-to-source voltage was realized by the RC series path of each MOSFET (such as R1 and C1 of Q1). The maximum drain-to-source on-state time of the MOSFET was determined by the RC time constant and the minimum gate threshold voltage of the MOSFET. In order to protect the MOSFET, Zener diodes (D5–D8) were used to prevent the MOSFET’s gate-to-source voltage from being too high. High break-voltage diodes (D1–D4) were used to specify the charge and discharge path of the CMUT.

### 2.2. Bias-Tee

In the front-end circuit of a CMUT, DC bias should be separated from AC signals to avoid mutual interference. There are two kinds of bias-tee [15,16], as shown in Figure 3. The bias-tee shown in Figure 3b loads DC bias and AC signals on the same plate of the CMUT, so it requires a large resistor at the DC channel and a capacitor at the AC channel to isolate the signals and bias. However, the large resistor at the DC channel introduces large thermal noise into the amplifier. Moreover, the capacitor at the AC channel introduces a pole in the frequency response of the amplifier, which decreases the bandwidth of the amplifier. In another bias-tee design, the DC bias and AC signals can be loaded on different plates of the CMUT (Figure 3a). As a result, there is no need to add any resistors and a capacitor to isolate the bias and signals, which improves the SNR and bandwidth of the front-end circuit. Therefore, the bias-tee design shown in Figure 3a was adopted in our front-end circuit.

### 2.3. T/R Switch

The single pole single throw (SPST) switches (ADG 5423, Analog Devices, Norwood, MA, USA), shown as S1 and S2 in Figure 1, were used as the T/R switch that controlled the on–off of the receiving and transmitting channels, respectively. The maximum supply voltage of ADG 5423 was ± 22 V, limiting the amplitude of AC excitation, as it cannot exceed 44 Vpp. The switches were controlled by the same FPGA, in which the low and high voltages of the control signal were 0 and 3.3 V, respectively. The sequence diagram of control is shown in Figure 4.

### 2.4. Amplifier

Both trans-impedance amplifiers (TIAs) and voltage (non-inverting) amplifiers can be used for amplifying CMUT output signals. Trans-impedance amplifiers are often used to convert the current signal produced by sensors into a voltage signal (e.g., as a front-end amplifier for photodiodes). A. Caronti et al. [17] pointed out that voltage (non-inverting) amplifiers have wider bandwidth, while trans-impedance amplifiers have higher SNR. High SNR is required for air-coupled CMUT, since the output signal of air-coupled CMUT is very small [18] (about a few microamps for our CMUT). Under 100 V DC bias, our air-coupled CMUT [13] only worked at resonant frequency of 217 kHz with a −3 dB fractional bandwidth of 15.7%, so the bandwidth requirement of the amplifier was not high. According to the above analysis, a trans-impedance amplifier was used in this paper.

In the trans-impedance amplifier design, input capacitance affects the amplifier’s frequency response, so a feedback capacitor is needed to compensate the amplifier’s frequency response [19]. The input capacitance mainly consists of the CMUT’s equivalent capacitance, common mode and differential mode input capacitance of the operational amplifier, and parasitic capacitance of the circuit board. In order to obtain the equivalent capacitance of the CMUT at the resonant frequency, the CMUT can simply be equivalent to an RC series circuit, and the equivalent capacitance can be calculated using the following equations:(1)$$\left\{ \begin{array}{l} A=\sqrt{R^{2}+\frac{1}{4\pi^{2}f^{2}C^{2}}}\\ \varphi =\arctan\left(-\frac{1}{2\pi fRC}\right) \end{array}\right.$$
where A is amplitude of impedance, φ is phase of impedance, R is equivalent resistance, C is equivalent capacitance and f is resonant frequency.

The impedance curve of our air-coupled CMUT, measured by an impedance analyzer (4294A, Agilent, Santa Clara, CA, USA), is shown in Figure 5, and the impedance at the resonant frequency of 217 kHz was 1.7 kΩ (amplitude) and −78.9° (phase). Calculated using Equation (1), the equivalent capacitance of our air-coupled CMUT was 0.44 nF. In general, the common mode and differential mode input capacitance of operational amplifiers and the parasitic capacitance of circuit boards are several hundred to several thousand fF, which can be ignored, compared to the equivalent capacitance of a CMUT. Therefore, the input capacitance was determined as 2.127 nF. In the trans-impedance amplifier in Figure 1, the −3 dB bandwidth $f_{-3dB}$ was
(2)$$f_{-3dB}=\sqrt{\frac{GBP}{2\pi R_{f}C_{i}}}$$
where GBP is the gain bandwidth product of the operational amplifier, $R_{f}$ is feedback resistance and $C_{i}$ is input capacitance.

In order to achieve a maximally flat second-order Butterworth frequency response, the feedback capacitance $C_{f}$ must satisfy the following equation:(3)$$\frac{1}{2\pi R_{f}C_{f}}=\sqrt{\frac{GBP}{4\pi R_{f}C_{i}}}$$

In this work, OPA657 (Texas Instruments, Dallas, TX, USA) was selected as the operational amplifier, whose gain bandwidth product is 1.6 GHz and input bias current is 2 pA. According to the output current of the CMUT (a few microamps), the trans-impedance gain $R_{f}$ was set to 100 kΩ. The bandwidth of −3 dB was calculated as 2.4 MHz, which met the bandwidth requirement of our CMUT. According to (3), the feedback capacitance was 0.94 pF, and the actual feedback capacitance was 1.0 pF. The simulation result of the amplifier’s frequency response by Multisim (National Instruments, Dallas, TX, USA) is shown in Figure 6.

## 3. Circuit Characterization

The integrated front-end circuit board for air-coupled CMUT was fabricated (Figure 7), followed by pulser characterization, burst-echo testing and SNR evaluation.

### 3.1. Pulser Characterization

In order to test the performance of the designed pulser, an oscilloscope (MSO5072, RIGOL, China) was used to acquire the output waveform of the pulser. The output waveform of 217 kHz 5-cycle 20 Vpp bipolar square-wave burst is shown in Figure 8. There was a small collapse after the square-wave burst, which was caused by the switch.

In order to test the transmitting waveform of the CMUT excited by our designed pulser, as well as to compare the influence of a square-wave burst and a sin-wave burst on the air-coupled CMUT’s transmission, a standard piezoelectric air-coupled ultrasonic transducer (NCG200-D25, Ultran Group, Hoboken, NJ, USA) was used to receive the ultrasonic signal transmitted by the air-coupled CMUT under 100 V DC bias, whose output signal was amplified by a commercial charge amplifier (Model.20/40/60 dB, Softland Times, Beijing, China) and subsequently acquired by an oscilloscope (MSO5072, RIGOL, Beijing, China). The square-wave burst was generated by the pulser reported in this paper, while the sin-wave burst was generated by a signal generator (SDG5162, SIGLENT, Shenzhen, China) and amplified by a power amplifier (Model2350, TEGAM, Cleveland, OH, USA). Both bursts had the same parameters in frequency (217 kHz), amplitude (20 Vpp) and number of cycles (five cycles). The center frequency of our CMUT was 217 kHz with −3 dB fractional bandwidth of 15.7%, and that of NCG200-D25 was 200 kHz with −3 dB fractional bandwidth of 40%, which fully covered the bandwidth of our CMUT, so the transmission signal of our CMUT could be totally acquired by the NCG200-D25. The experimental results shown in Figure 9 indicate that the maximal amplitude of the transmitted signal excited by the square-wave burst was improved by about 20%, compared to the sin-wave burst. In addition, the waveform in Figure 9 proves that the small collapse of the square-wave burst did not have a significant impact on the CMUT’s transmitting waveform.

### 3.2. Burst-Echo Characterization

The burst-echo experiment of the air-coupled CMUT was carried out to test the performance of the front-end circuit. A plastic plate was placed in front of the CMUT to reflect an ultrasonic wave during the test. The distance between the CMUT and plastic plate was 60 mm, and the CMUT was excited by a square-wave burst with the frequency of 217 kHz, amplitude of 20 Vpp and five cycles. The experimental result shown in Figure 10 indicates that the front-end circuit realized the function of burst-echo applications. When the receiving channel was closed, the leakage current of AC excitation still could pass through, but these currents were too small to harm the operational amplifier.

### 3.3. SNR Characterization

In order to evaluate the noise characteristics of the front-end circuit board, the effective values of noise voltage and signal voltage were tested and calculated using the following equation:(4)$$U=\sqrt{\frac{1}{T}\int_{0}^{T}{\left[u(t)\right]}^{2}dt}$$

The effective value of noise voltage was obtained under the condition of zero input, mainly considering the noise introduced by the amplifier, switches, CMUT and DC offset. The effective value of signal voltage was obtained at the reflector distance where the echo signal had maximum amplitude. It showed that the noise came mainly from the switch, and the CMUT could slightly reduce the noise, which might have been because the CMUT capacitance had a certain filtering effect on noise (Table 1). SNR of the front-end circuit board (including the CMUT) was calculated to be 22.94 dB, via the following equation:(5)$$SNR=20\log_{10}\frac{U_{S}}{U_{N}}$$
where $U_{S}$ is the effective value of the echo signal, and $U_{N}$ is the effective value of overall noise.

## 4. Imaging

### 4.1. Imaging Method

The burst-echo imaging of air-coupled CMUTs, based on time of flight, was conducted using 2D scanning with a step of 1 mm in both directions (Figure 11). Imaging distance between the CMUT and the target was 60 mm. The echo signal from a “T” imaging target with a convex height of 5 mm (Figure 12) was acquired by using an acquisition card (ADQ12, Teledyne SP Devices, Linköping, Sweden). Ultrasonic waves rarely penetrated the interior of the “T” imaging target. A 16-element air-coupled CMUT array (15.8 × 15.8 mm, as shown in Figure 13) [13] with a center frequency of 217 kHz was used under 100 V DC bias. The object was carefully aligned with the transducer to avoid tilting the image surface. A 20 Vpp five-cycle square-wave burst, produced by the designed front-end circuit, was used as AC excitation, with the frequency of 217 kHz.

The surface morphology (C-scan) was reconstructed using the reflected waves at different locations. The delay time between the reflected waves was calculated using cross-correlation [20]. In order to eliminate the imaging error caused by surface flatness, material uniformity and edge integration of the 3D-printed object, the image was processed using adaptive threshold filtering, where the threshold was decided by the numerical values and distribution of adjacent points in a 3 × 3 matrix. The image was interpolated linearly five times in both directions to improve imaging sharpness.

### 4.2. Imaging Result

The imaging result in Figure 14 shows a clear “T” surface morphology and convex height of 5 mm, which agreed with the shape well. The alignment result between the object and the transducer was good, and the distance difference between the four corners was less than 0.1 mm. However, the imaging result is irregular in Figure 14a, which is mainly due to the lack of 3D printing quality, including flatness, uniformity and edge integration. After adaptive threshold filtering and linear interpolation, the outliers could be eliminated and the edge smoothed (Figure 14b). The cross-section plot of Figure 14b (at y = 44 mm) is shown in Figure 15. There was a process of gradually increasing to the height of 5 mm and gradually decreasing to the height of 0 mm. The lateral length at a height of 5 mm (±1%) was 7 mm, which was smaller than the actual length of 8 mm. The difference of lateral length at a height of 5 mm was mainly caused by the impact of the CMUT’s beam width and the algorithm of imaging reconstruction. The time difference between the reflected waves was calculated using cross-correlation. The cross-correlation results related to both the start time and the waveform of the echo signal, so better lateral imaging resolution was achieved, compared to the transducer’s beam width. The lateral length of 8 mm was acquired, as shown in Figure 15, when the height was 4.2/5 mm (±1%).

### 4.3. Imaging Range

The maximum imaging range was mainly affected by transmission energy and receiving sensitivity of the CMUT, while the nearest imaging range was affected mainly by the length of the near field and signal tailing of the CMUT.

In order to determine the imaging range of the CMUT, we obtained the relationship between the SNR of the echo signal and the imaging distance by an experiment. The experimental result is shown in Figure 16. The distance at the maximum SNR was approximately the focus of the CMUT and the length of the near field, which was 52 mm. The length of burst and signal tailing was about 210 μs, as shown in Figure 10, and the corresponding distance was 35.7 mm, as calculated by the following equation:(6)$$Distance=\frac{c\times Time}{2}$$
where c is the velocity of ultrasound, which is 340 m/s.

Nearest imaging range should be longer than both the length of the near field and signal tailing, which was 52 mm. The maximum imaging range was where the SNR of echo signal reduced to 6 dB, which was 273 mm.

It should be noted that the maximum imaging range of 273 mm might be a limitation in some of the CMUTs’ imaging applications in air. Within this range, the single-element sensor or an array has potential to be used for noncontact operations of electronic devices (gesture recognition), anticollision systems of robotic vacuums (ultrasonic ranging), counting and selecting pieces on a conveyor (ultrasound imaging) and so on.

The maximum imaging range could be enlarged by improving the CMUTs’ sensitivity, reducing its center frequency, improving excitation voltage, improving excitation cycle numbers and improving the system’s SNR. However, reducing the CMUTs’ center frequency would decrease imaging resolution, and improving excitation cycle numbers would increase the length of transmission signal tailing, which would increase the nearest imaging range. Therefore, improving the CMUTs’ sensitivity, excitation voltage and the system’s SNR might be a good method to increase the maximum imaging range in further studies.

## 5. Conclusions

In order to achieve air-coupled CMUT’s application with the same transmission and reception elements for the first time, a dedicated and integrated front-end circuit, controlled by an FPGA, was designed, fabricated and tested, followed by burst-echo imaging.

The designed front-end circuit board controlled by an FPGA consisted of four parts: an on-board pulser, a bias-tee, a T/R switch and an amplifier, and the measured SNR was 22.94 dB, which was high enough to perform burst-echo imaging. Using the designed front-end circuit board, burst-echo imaging of the air-coupled CMUT, based on time of flight, was carried out. The imaging result could represent the surface morphology and the three-dimensional form factor of the imaging target. Finally, burst-echo imaging accuracy and imaging range of our air-coupled CMUT imaging system were analyzed, and the imaging range was found to be from 52 to 273 mm.

This work demonstrated the feasibility of air-coupled CMUT imaging using the same transmission and reception elements, suggesting good potential in ultrasound imaging and gesture recognition applications in the future.

## Figures and Tables

**Figure 1 sensors-20-06128-f001:**
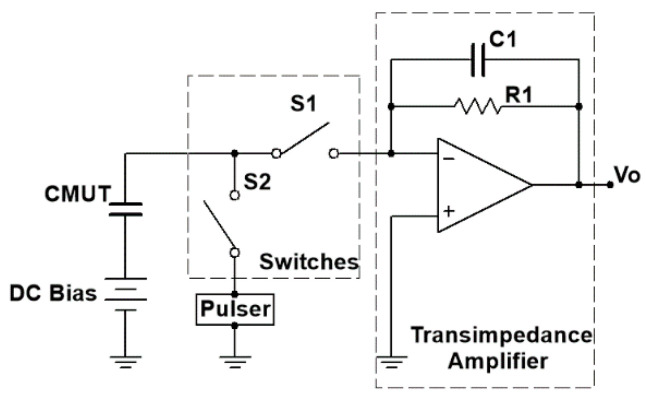
Circuit structure of the integrated front-end circuit.

**Figure 2 sensors-20-06128-f002:**
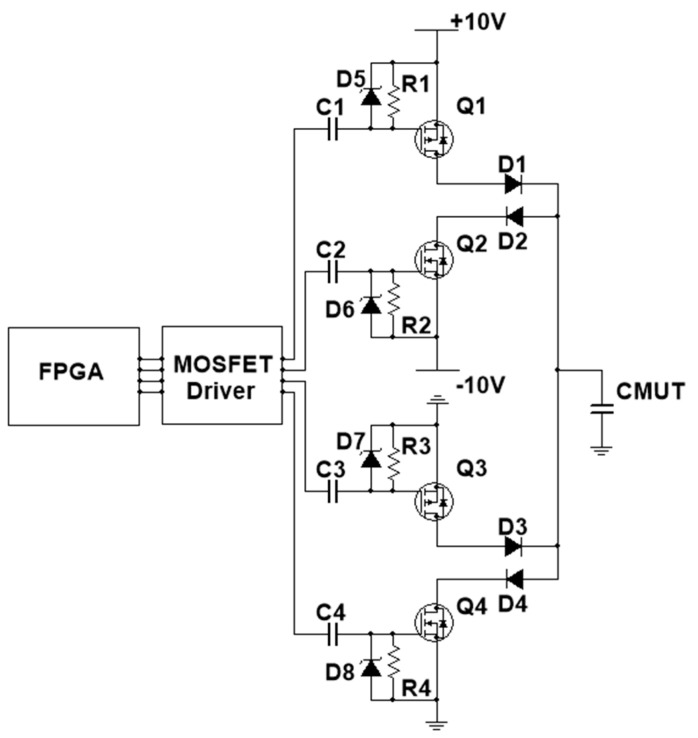
Circuit design of the bipolar square-wave pulser.

**Figure 3 sensors-20-06128-f003:**
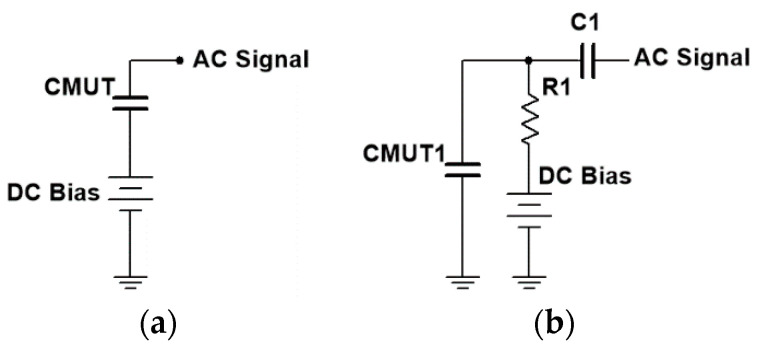
Two forms of the bias-tee design: (**a**) load DC bias and AC signals on different plates of the capacitive micromachined ultrasonic transducer (CMUT); (**b**) load DC bias and AC signals on the same plate of the CMUT.

**Figure 4 sensors-20-06128-f004:**
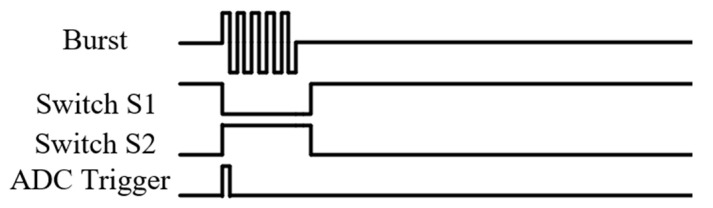
Sequence diagram of control.

**Figure 5 sensors-20-06128-f005:**
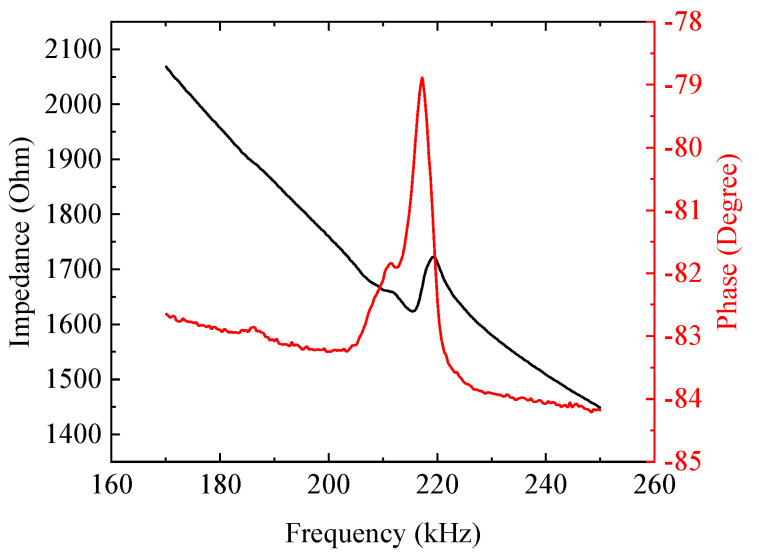
Impedance curve of our air-coupled CMUT.

**Figure 6 sensors-20-06128-f006:**
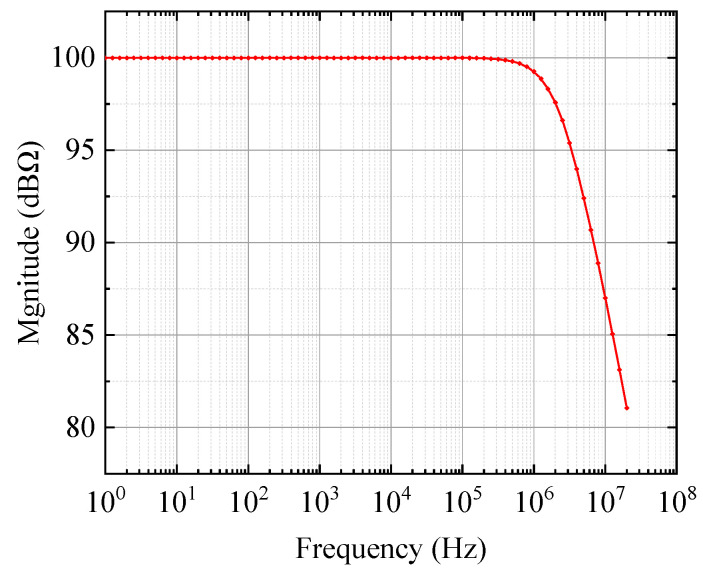
Simulation result of the amplifier’s frequency response.

**Figure 7 sensors-20-06128-f007:**
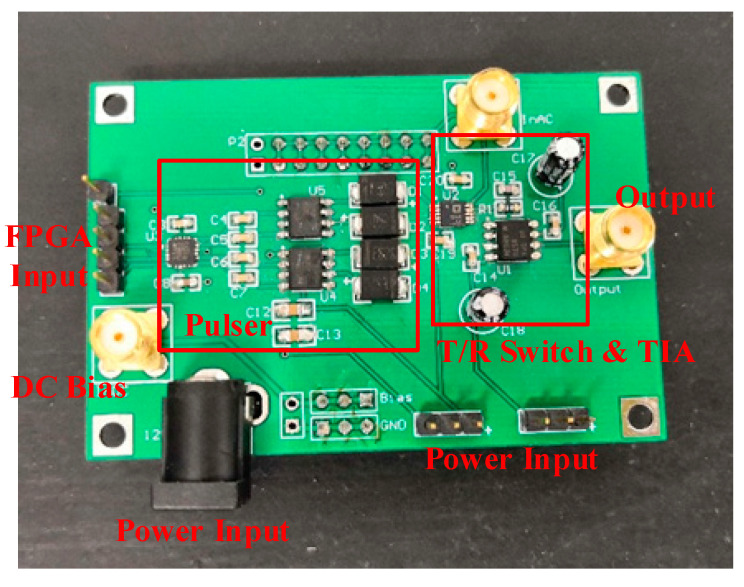
Photograph of the integrated front-end circuit board.

**Figure 8 sensors-20-06128-f008:**
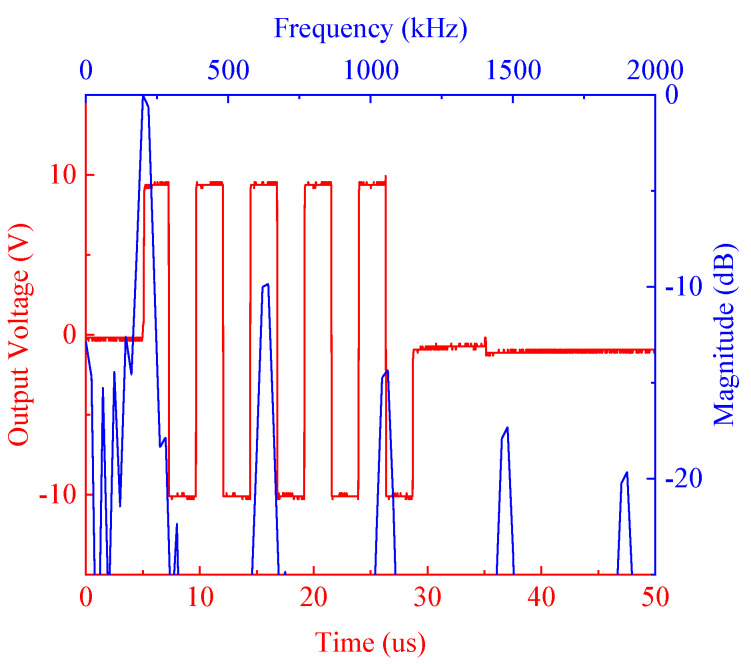
Output waveform of the pulser and its spectrum.

**Figure 9 sensors-20-06128-f009:**
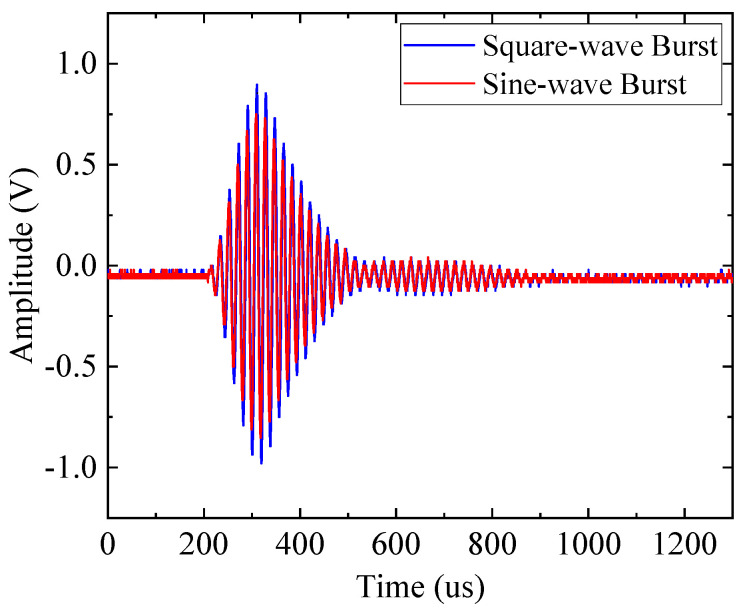
Transmitting signal of the CMUT excited by the square-wave and sin-wave bursts.

**Figure 10 sensors-20-06128-f010:**
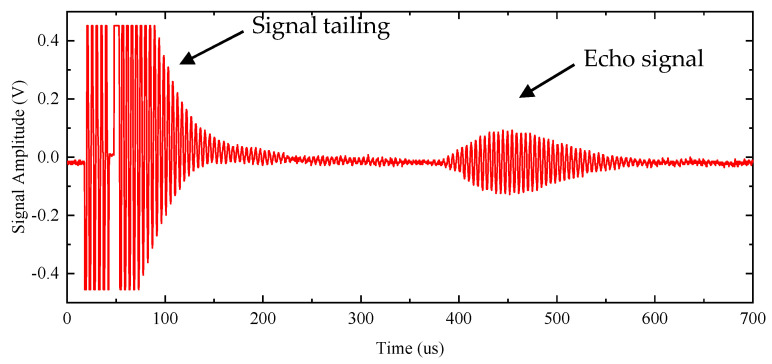
Output signal (echo) of the front-end circuit under five-cycle burst excitation.

**Figure 11 sensors-20-06128-f011:**
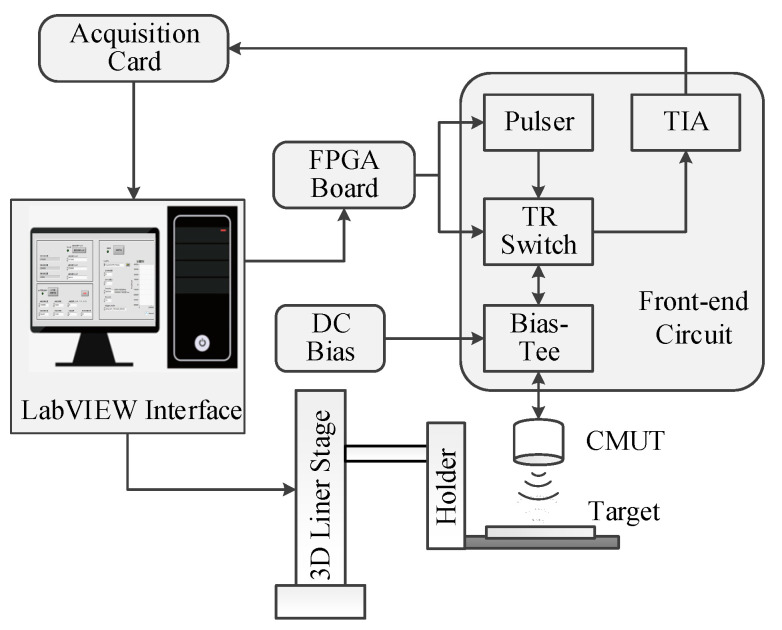
Schematic of burst-echo imaging.

**Figure 12 sensors-20-06128-f012:**
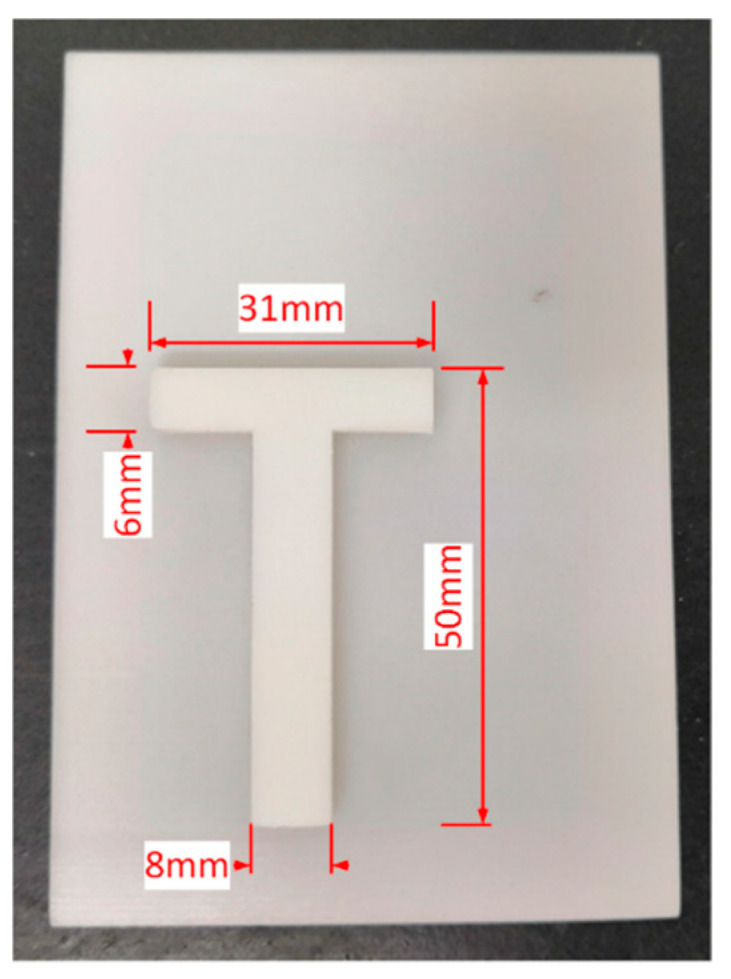
3D-printed “T” shape imaging target.

**Figure 13 sensors-20-06128-f013:**
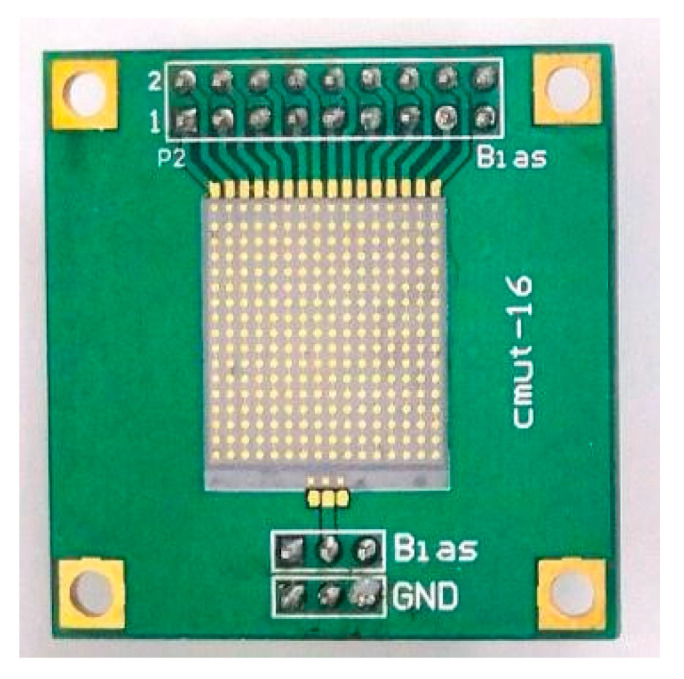
Picture of the CMUT.

**Figure 14 sensors-20-06128-f014:**
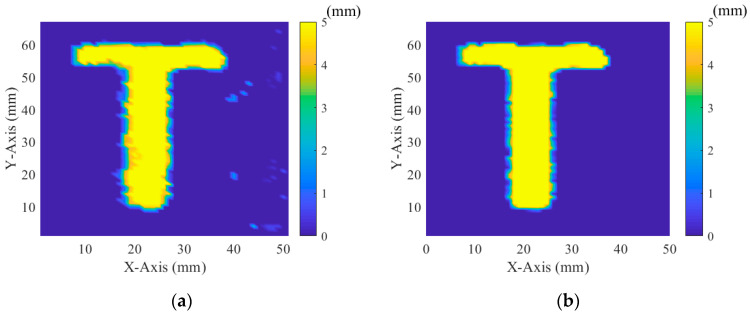
Imaging results of the “T” imaging target: (**a**) imaging result before the imaging process; (**b**) imaging result after the imaging process.

**Figure 15 sensors-20-06128-f015:**
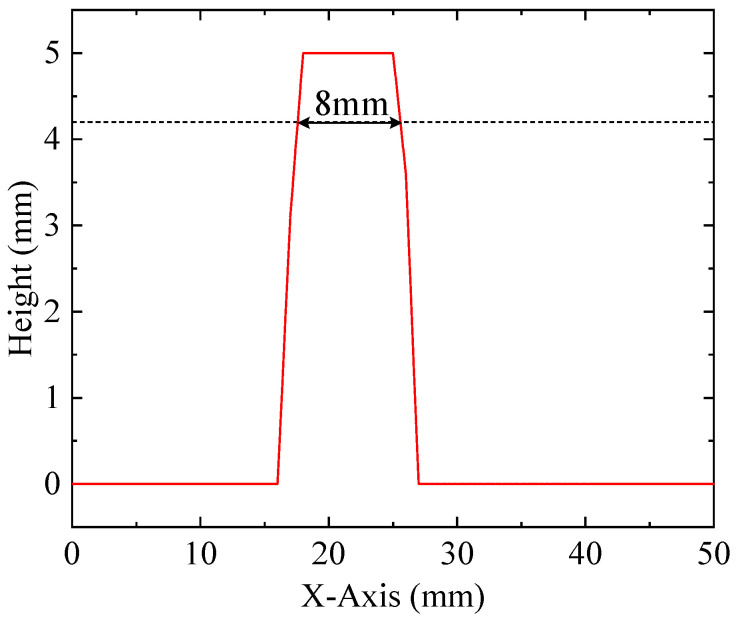
Cross-section plot of the imaged object (at y = 44 mm).

**Figure 16 sensors-20-06128-f016:**
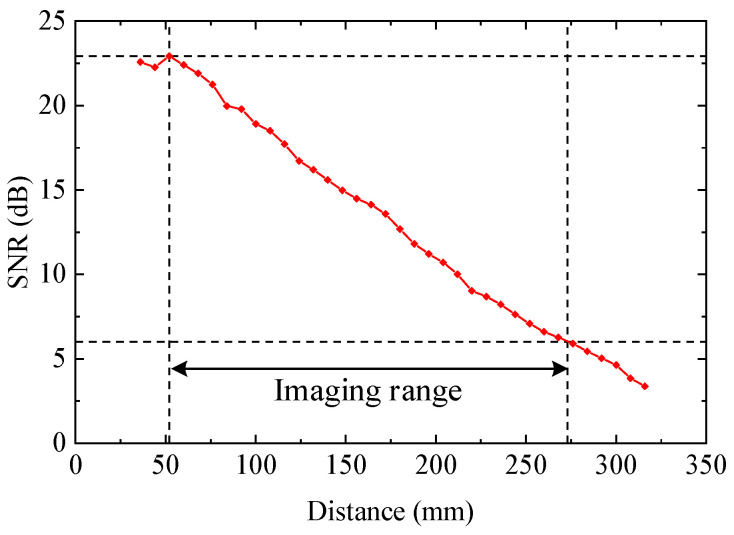
Relationship between the SNR of the echo signal and the imaging distance.

**Table 1 sensors-20-06128-t001:** Effective values of noise and signal.

Noise or Signal	Effective Value (mV)
Noise of amplifier	0.70
Noise of amplifier and switches	3.80
Noise of amplifier, switches and CMUT	3.40
Noise of amplifier, switches, CMUT and DC bias	3.80
Signal of echo	53.30

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
