# Peer review of "An Integrated Front-end Circuit Board for Air-Coupled CMUT Burst-Echo Imaging"

_sensors, 2020, doi:10.3390/s20216128_

Round 1

Reviewer 1 Report

Review of: An Integrated Front-end Circuit Board for Air-coupled CMUT Burst-echo Imaging.

By: Lei Ye et al.

The work is of interest and is very well presented, with precision, clarity and a pretty good organization. It will be of interest for the journal readers. I present some suggestions below for authors consideration.

My main concern is about the hypothesis, repeated several times by the authors, that there are no pulse-echo applications of CMUT air-coupled transducers. However there are a number of works where CMUT air-coupled transducers have been used and characterized in pulse-echo mode. Therefore, authors should more precisely define what is the difference between what they propose and what have been previously done. In its present formulation I find it far too general to be accepted.

Abstract: Define all acronyms before used for the first time

Page 1, line 36:  “doesn’t use” instead of “does not used”.

Page 1, line 40: “conflicts with” instead of “is conflict to”

Lines 160-171. I understand that the purpose of this paragraph together with figure 9 is to show the improvement obtained with the square burst compared with the sine burst, provided they have the same amplitude. However this comparison is not well balanced because what really matter is the energy provided within the transducer frequency band (and harmonics). As the spectral distribution of the square and sine bursts is different, this conclusion is not fully demonstrated. In addition, there could be potential differences in the impedance matching of the two different pulsers used with the CMUT transducer. Alternatively, this could have been done by comparing the ratio of the amplitude of the FFT of the received signal to the amplitude of the FFT of the signal applied to transducer terminals for both cases. However, this would be valid only if all received energy is within the bandwidth of the receiver transducer. Therefore, the frequency band of the Ultran transducer and the CMUT transducer must also be provided.

Line 164: Specify the brand and model of the commercial charge amplifier used. Is it the one mentioned in line 140?

Section 4.1. To understand the images obtained further details about the transducer are required as effective aperture geometry and size, if the transducer it is flat or focused.

Line 203. Why is the 5 cycles selected? If transducer bandwidth is shown this could help to solve this question.

Line 208. Please specify the threshold filtering method employed.

Lines 205-208. In this type of measurements it is quite normal to find that there is a small but measurable misalignment between transducer and object, so that imaged surface is eventually slightly tilted so that distance between transducer and object is no always equal. This can be easily corrected during the signal processing. In Figure 13, no such effect is observed. I wonder if authors were extremely precise in doing so or this has been eliminated during the threshold filtering.

Line 218. The lateral resolution of the CMUT should be specified.

Figure 13. It would be good to see the raw image along with the processed one so that the reader can have an estimation of what is the contribution of the transducer design and performance and how important is the processing.

Figure 14. May be the agreement between estimated width of the T and the actual value is far too good, isn’t it? At 215 kHz the wavelength in air is about 1.6 mm and scan step is 1 mm.

Figure 15. Maximum range of 273 mm is rather limited for imaging in air, gesture recognition and human machine interaction. Authors should comment about this limitation and discuss about potential improvements that could enlarge the range (transducer sensitivity optimization,  frequency reduction, higher voltage excitation, noise reduction, signal processing??)

Reviewer 2 Report

The paper is technically sound and should be accepted for publication. Few points need to be addressed: 1) Authors should add a picture of Burst-echo characterization setup, please calculated the wave velocity from the echo signal. 2) Please discuss capabilities of the CMUT transducer for thickness measurement and internal defect characterization.

Reviewer 3 Report

CMUT has been widely used in ultrasonic imaging, therapy and gesture recognition. The front-end circuit is used to drive CMUTs or to capture the signals received by CMUTs. This paper devised an integrated front-end circuit for air-coupled CMUT to realize burst-echo imaging with the same elements in transmission and reception. This is helpful to reduce the size of the CMUT array. The authors fabricated the integrated front-end circuit and tested the performance for burst-echo imaging using their previous CMUT chips. The results demonstrated the feasibility of the designed circuits for burst-echo imaging. This paper can be accepted for publication on this journal after major revisions.

  1. The author should check the grammar errors and refine the English writing throughout the entire manuscript.

For instance:

Page 1, line 25; “Comparing to traditional piezoelectric ultrasonic transducers” should be “compared to...”

Page 1, line 30; “which -6dB bandwidth was improved from 1% to 2.5% for” should be “ in which the -6dB bandwidth... or the -6dB bandwidth of which...”

Page 1, line 36; “However, previous applications of air-coupled CMUTs does not used” should be “... does not use”

Errors also can be seen in “which is conflict to its merits of miniaturization and...” (Page 1, line 40)

...

There are a lot of these kind of grammar errors, please check the entire manuscript carefully and make thorough modifications on the English writing.

  1. About the imaging results, “The lateral length at a height of 5 mm is 7 mm, which is slightly smaller than the actual length of 8 mm”. What is the reason for the difference?
  2. It’s better to clearly point out each functional sections on the integrated front-end circuit board. For example, which section is T/R switch? Which section is Bias-Tee circuit?...
  3. It’s suggested to show the picture, or schematic of the fabricated CMUT chips somewhere, or in appendix section, or supporting documents.

Reviewer 4 Report

Comment

Dear editor:

Thank you for inviting me to evaluate the article titled “An Integrated Front-end Circuit Board for Air-coupled CMUT Burst-echo Imaging”. In this paper, the authors report a front-end circuit based on FPGA to establish an ultrasound imaging system for capacitive micromachined ultrasound transducers. The authors have done an excellent technical job. This is definitely worthy of consideration. However there are some technical and non-technical issues that need further clarification before this paper can be accepted.

Reject comments:

  1. This article has the problem of insufficient innovation in the research of integrated circuits.
  2. Figure 10 in the article, Is this echo signal under burst excitation the actual test result?
  3. In this paper, the circuit design of the imaging system is not highly integrated.If the circuit integration can be improved, then this research will be more valuable.

Round 2

Reviewer 3 Report

The  manuscript has been significantly improved over last version. It can be accepted for publication.

Author Response

Thank you very much for the suggestions.

We proofread the paper carefully and grammar errors have been corrected.

On Page 1, Line 44. “leading a low signal-to-noise ratio (SNR)” has been modified to “leading to a low signal-to-noise ratio (SNR)”.

On Page 2, Line 63. “designing amplifying circuit” has been modified to “designing an amplifier”.

On Page 2, Line 63. “the amplifier circuit” has been modified to “the amplifier”.

On Page 3, Line 85. “from AC signal” has been modified to “from AC signals”.

On Page 3, Line 91. “In the another bias-tee design” has been modified to “In another bias-tee design”.

On Page 3, Line 105. “Sequence diagram of control” has been modified to “The sequence diagram of control”.

On Page 4, Line 112. “into voltage signal” has been modified to “into a voltage signal”.

On Page 4, Line 124. “a RC series circuit” has been modified to “an RC series circuit”.

On Page 5, Line 146. “Simulation result of the amplifier’s frequency response by Multisim (National Instruments, US) are shown in Figure 6.” has been modified to “The simulation result of the amplifier’s frequency response by Multisim (National Instruments, US) is shown in Figure 6.”.

On Page 7, Line 188. “the function of burst-echo application” has been modified to “the function of burst-echo applications”.

On Page 8, Line 208. “in both direction” has been modified to “in both directions”.

On Page 8, Line 217. “In order to eliminates” has been modified to “In order to eliminate”.

On Page 8, Line 220. “in 3 × 3 matrix” has been modified to “in a 3 × 3 matrix”.

On Page 11, Line 280. “was design” has been modified to “was designed”.

All the revisions have been highlighted in the latest version.

Reviewer 4 Report

Dear editor:

Thank you for inviting me to evaluate the article titled “An Integrated Front-end Circuit Board for Air-coupled CMUT Burst-echo Imaging”. In this paper, the authors report a front-end circuit based on FPGA to establish an ultrasound imaging system for capacitive micromachined ultrasound transducers. The authors have done an excellent technical job. This is definitely worthy of consideration.Since they have addressed all my questions, I am glad to recommend the acceptance of this manuscript.

Author Response

(The authors gave the same response as above.)
